# Interdependence of Molecular Lesions That Drive Uveal Melanoma Metastasis

**DOI:** 10.3390/ijms242115602

**Published:** 2023-10-26

**Authors:** Francesco Reggiani, Marianna Ambrosio, Michela Croce, Enrica Teresa Tanda, Francesco Spagnolo, Edoardo Raposio, Mariangela Petito, Zeinab El Rashed, Alessandra Forlani, Ulrich Pfeffer, Adriana Agnese Amaro

**Affiliations:** 1Laboratory of Gene Expression Regulation, IRCCS Ospedale Policlinico San Martino, 16132 Genova, Italy; 2Department of Experimental Medicine (DIMES), University of Genova, Via Leon Battista Alberti, 16132 Genova, Italy; 3Biotherapies, IRCCS Ospedale Policlinico San Martino, 16132 Genova, Italy; 4Skin Cancer Unit, IRCCS Ospedale Policlinico San Martino, 16132 Genova, Italy; 5Department of Internal Medicine and Medical Specialties, University of Genova, Viale Benedetto XV, 16132 Genova, Italy; 6Department of Surgical Sciences and Integrated Diagnostics (DISC), University of Genova, 16132 Genova, Italy; 7Plastic Surgery Division, Department of Surgical Sciences and Integrated Diagnostics (DISC), University of Genova, 16132 Genova, Italy

**Keywords:** uveal melanoma, metastases, gene expression, BAP1, CNA, tumor evolution

## Abstract

The metastatic risk of uveal melanoma (UM) is defined by a limited number of molecular lesions, somatic mutations (SF3B1 and BAP1), and copy number alterations (CNA): monosomy of chromosome 3 (M3), chr8q gain (8q), chr6p gain (6p), yet the sequence of events is not clear. We analyzed data from three datasets (TCGA-UVM, GSE27831, GSE51880) with information regarding M3, 8q, 6p, SF3B1, and BAP1 status. We confirm that BAP1 mutations are always associated with M3 in high-risk patients. All other features (6p, 8q, M3, SF3B1 mutation) were present independently from each other. Chr8q gain was frequently associated with chr3 disomy. Hierarchical clustering of gene expression data of samples with different binary combinations of aggressivity factors shows that patients with 8q|M3, BAP1|M3 form one cluster enriched in samples that developed metastases. Patients with 6p combined with either 8q or SF3B1 are mainly represented in the other, low-risk cluster. Several gene expression events that show a non-significant association with outcome when considering single features become significant when analyzing combinations of risk features indicating additive action. The independence of risk factors is consistent with a random risk model of UM metastasis without an obligatory sequence.

## 1. Introduction

Uveal melanoma (UM) is the most common primary intraocular malignancy in adults and accounts for 80% of all non-cutaneous melanomas [1]. Although the molecular characteristics of the primary tumor are thoroughly known, metastatic disease remains incurable with very low 5-year survival that has not changed for decades [2]. Despite low risk of local recurrence, metastatic disease in the liver, develops in 25% to 30% of patients within 5 years and in approximately 50% of patients within 10 years. No adjuvant treatments have been identified to delay metastasis, and the median survival time after diagnosis of metastatic UM is 6–9 months [3]. Recently Tebentafusp has been approved by FDA for metastatic uveal melanoma treatment, with an OS benefit observed in treated patients [4]. 

Uveal melanoma and cutaneous melanoma (CM) arise from melanocytes that originate from common ancestors, namely the neural crest-derived melanoblasts. Both show a higher incidence in subjects with a fair complexion. Despite the common origin and shared risk factors, UM and CM show very distinct characteristics. The prevalent mutation types in the two melanomas are C > T transitions, yet they differ for the prevalent mutational signature, linked to exposure to ultraviolet rays for CM but not for UM [5]. No etiological agents that increase the risk of UM are known; UV rays are almost entirely filtered by the lens and vitreous body of the eye and do not reach the uvea. UM and CM carry different tumor initiating-mutations, GNAQ, GNA11, CYSLTR2, PLCB4 in UM and BRAF, NRAS, NF1 in CM, and show a different frequency of mutations (mutational burden), very high for CM (>10 mutations per megabase [6]) and very low for UM (0.5 mutations per megabase [7]). These features also determine a different clinical course that has dramatically improved for CM through the introduction of targeted therapy and immune checkpoint inhibitors [8] that show only very limited benefit for UM [9]. A recent work of Newell and coauthors has shown the presence of 6p and 8q gain, typical of UM, in a set of samples from different melanoma subtypes as cutaneous, acral and mucosal [10]. Interestingly, several UM samples with mutated p53 were described for the first time [10].

Metastatic risk is determined by cytogenetic alterations; mainly monosomy of chromosome 3 (M3), amplification of chr8q, and inversely, by chr6p gain, by somatic mutations in BAP1 (high risk), SF3B1 (intermediate risk, metastasis with long latency), and EIF1AX (low risk), by cell shape (spindle or epithelioid cells), tumor thickness, basal diameter and by gene expression profile (class 1, class 2 signature) [11]. The initiating mutation in GNA11 also determines a more aggressive development of the tumor as compared to GNAQ mutations [12].

BAP1 was initially identified as a protein interacting with the tumor suppressor gene BRCA1, which is involved in homology-directed DNA repair. The loss of BAP1 hinders homology-directed DNA repair, forcing cells to rely on the more error-prone non-homologous end joining (NHEJ) [13]. Unexpectedly, the mutational load in UM is noticeably lower compared to other cancer types, indicating that BAP1 deletion in UM does not severely impact DNA damage repair mechanisms [13]. It is therefore likely that other functions of BAP1 determine the metastatic risk. In fact, its role in polycomb repressive complex 1 (PRC1) has been linked to the metastatic potential [14]. Recent work has shown that BAP1 biallelic inactivation could promote metastasis development by enhancing immune evasion of the tumor through inhibition of the immune response [15], consistent with the limited response of UM to immunotherapy [9]. The splicing factor 3b, encoded by the SF3B1 gene, is responsible for appropriate branchpoint detection during pre-mRNA splicing. SF3B1 mutations in UM lead to alternative splicing at the 3′ end of exon borders and can cause aberrant splicing. Incorrectly spliced transcripts may be translated into specific, abnormal proteins, or their expression may be lost due to nonsense-mediated RNA decay [16].

For the sake of prognostication, molecular events are grouped into risk classes. The most accepted molecular risk model has been provided by the analysis of 80 UM cases by integrative genomics, which provided whole genome copy number alteration (CNA), gene expression, and DNA-methylation data. Each of these domains identifies four risk classes that roughly correspond to the disomy of chr3 with mutation of EIF1AX (class 1) or SF3B1 (class 2) and chr3 monosomy and BAP1 mutation without (class 3) or with (class 4) chr8q amplification [13].

The Robertson classification also significantly correlates with prognosis in other clinical UM datasets [17] yet in the clinics, more complex predictors are applied. LUMPO 3 (Liverpool Uveal Melanoma Prognosticator Online), for example, estimates the absolute risk of metastatic death from choroidal melanoma by combining features such as chromosome 3 loss, 8q gain, and Tumor Node Metastasis/American Joint Committee on Cancer (TNM/AJCC) staging [18,19]. Recent work has analyzed the effect of CNA on a large cohort of around one thousand UM samples: this study extended the Robertson classification by adding 16q and 1p deletion to high-risk class definition [3]. The high number of samples in this study enlightened the presence of a rare high-risk class (5), defined by chromosome 3 monosomy, 8q amplification, and 16q or 1p loss. 16q loss has previously been associated with metastastatic risk [20]. Chr16q contains a relevant proportion of tumor suppressors [21] but is generally a rare genomic alteration [3]. Other events, such as chr1q gain, are detected in metastases but rarely so in primary tumors [22,23]. CNAs could also be induced by a different selective pressure acting in new environments, as the observed increase of chr8q gains in metastatic versus primary tumors [3,24]. The gain of 6p has been previously reported as a protective factor [25], other authors have suggested a potential role of this CNA on UM malignancy, as HLA class I resides on chromosome 6. Overexpression of the antigen-presenting machinery could protect the tumor from Natural Killer cells [26].

Molecular prognostic classes, however, do not necessarily correspond to the sequence at the time of acquisition of molecular lesions that drive tumor evolution. Gene expression and, to a certain extent, DNA methylation are helpful for prognostication. However, they constitute intermediate phenotypes secondary to stable molecular lesions (somatic mutations and CNA) [12,13]. 

BAP1 mutations and chr3 monosomy are central to the metastatic risk [25]. Somatic mutation of the tumor suppressor gene BAP1, located on chr3, confers a selective growth advantage only in the presence of Chr3 monosomy leading to the complete loss of functional BAP1 expression. Loss of BAP1 also confers a gene expression profile associated with high metastatic risk [27,28]. The potential interdependence of the other molecular events driving UM tumorigenesis and metastasis is a matter of discussion.

BAP1 mutations are early events in tumorigenesis, likely occurring when the tumor still consists of a few malignant cells. Nevertheless, they do not imply the immediate departure of metastases [29]. After the cell has acquired a mutation in the GPCR signaling pathway that drives proliferation, BAP1 expression is lost and cells invade the surrounding tissue and migrate to choroidal vessels for metastasis. 

UM metastases do not show consistent acquisitions of additional molecular lesions concerning the primary tumors. Metastasis driver events seem to occur early during tumor development [30]. 

The tumor evolution model of successive molecular alterations that confer a growth advantage and accumulate over time has recently been challenged by the punctuated equilibrium or “big bang” model that predicts an original phase of genomic instability followed by the outgrowth of stabilized clones carrying molecular lesions that confer a selective growth advantage [2,31,32]. 

Here we try to systematically approach the interdependence of the molecular lesions of UM that determine metastasis thereby identifying the gene expression proxy of these associations. Identifying the molecular basis underlying UM evolution can provide insight into the molecular pathogenesis of UM calling for appropriate patient stratification in clinical trials of new drugs.

## 2. Results

### 2.1. UM Genomic Features and Metastases

We analyzed the frequency and the co-occurrence of five risk factors: BAP1 and SF3B1 mutations, M3, chr8q, and chr6p gains. We did not consider EIF1AX mutations since they are not associated with metastatic risk likely co-driving tumor initiation but not progression and we also excluded GNA11 as a risk factor which, in comparison to the other risk factors, only contributes a minor risk elevation [12].

Our analysis shows that chr8q gain is the most frequent molecular lesion encountered, present in 82 of 113 cases, followed by monosomy of chr3 observed in 60 cases and chr6p gain in 53 cases. BAP1 and SF3B1 mutations are present in 44 and 20 cases, respectively. Figure 1 shows the association of these features. Chr8q gain is mainly associated with chr3 monosomy but can occur separately and it occurs in the presence or absence of any of the other features. There are three cases with chr8q gain as an isolated feature. Any of the other features, namely chr3 monosomy, chr6p gain, BAP1, and SF3B1 mutations can occur in the presence or absence of any of the other features except for BAP1 mutation that is only encountered in cases with chr3 monosomy (Figure 1), consistent with its tumor suppressor gene function.

This picture dramatically changes when only the 41 cases that developed metastases within the time of follow-up are considered. Metastatic UM generally shows the presence of more than one molecular risk factor. The most numerous group among metastatic patients is characterized by 8q, M3, and BAP1 (15), followed by samples with also 6p gain (10), only M3 and 8q gain (6), and 6 samples with at least a mutation on SF3B1. The only feature that occurs in isolation in the absence of any of the other features, though in a single case, is chr3 monosomy. All features can occur in the presence or absence of any other feature again with the exception of the BAP1 mutation that is obligatorily associated with chr3 monosomy (Figure 2a). Cases that did not develop metastases within the time of follow-up show associations of molecular lesions similar to the whole samples as described above (Figure 2b).

Eight cases show none of the five molecular lesions analyzed. Seven of these, in the absence of metastasis driver lesions, did not develop metastases but one case did. This case probably has other, uncommon pro-metastatic lesions (Figure 2a).

### 2.2. Association of Pairs of Risk Factors with Gene Expression

To evaluate the effect of combined genomic events on gene expression and their effect on metastasis development, we compared gene expression profiles of samples with binary combinations of molecular risk factors with those samples where the specific combination was absent. Genes that are differentially expressed in any risk factor combination are reported in a single heatmap that also shows clinical and molecular sample annotations (Figure 3). 

Groups with 6p gain are clustered in the right part of the heatmap, which contains most samples without metastases. This was expected since a 6p gain is an indicator of a better prognosis. In general, monosomy of chromosome 3 is the dominant factor in patient prognosis: all combinations of M3 (with BAP1, 8q) led to the worst prognosis. In particular, the protective effect of 6p is evident in disomic patients (Appendix A) compared to monosomic samples (Figure 4a,b). 

In particular, 10 out of 12 M3 patients with 6p, 8q gain, and BAP1 have developed metastasis during follow-up, while progression was observed in only a limited number of disomic patients with SF3B1 (Figure 2a). In general, survival curves of SF3B1 mutated samples show a better prognosis compared to the other groups unless the mutation occurs in combination with M3 (Figure 4). SF3B1 mutations are considered a risk factor for delayed metastasis that develops up to 15 years later [33]. Gene ontology enrichment of differentially expressed genes shows enrichment of genes involved in the biological processes of second-messenger-mediated signaling and cyclic-nucleotide-mediated signaling: in UM, consistent with the constitutive activation of G protein signaling due to mutations in GNAQ, GNA11 genes [5,12,34,35]. To check if differentially expressed genes (DEG) were exclusively associated with a particular genomic feature (such as M3 or 8q) and not with the combination of features (M3/8q), we again performed DEG analysis comparing single factors (such as M3 vs. all) and we plotted the genes that were differentially expressed in more than one comparison (Figure 5).

Two clusters of patients can be observed (Figure 5): one mostly characterized by 8q, M3, and BAP1 mutations, comprising most of the metastatic patients, while the other group has a greater number of 6p gain events: in particular 8q gain is frequently associated with 6p gain, as in Figure 2 (Appendix A). The relation between gene expression and methylation in elements of Appendix A has been reported in Figure 6 with a scatterplot, as previously implemented in [36]. 

Each point coordinate is defined as the log fold change of the mean expression or gene methylation values in the high-risk samples over the low-risk ones. For most of the genes, we observe a concordance between the considered genomic domains (such as high expression and low methylation or the opposite), except for 11 items, 4 of them in regions covered by CNA (gray triangles). HCP5 is the gene with the lowest logFC methylation values, it maps on the 6p chromosome, and it has been shown to have a role in autoimmune diseases as well as promoting proliferation and metastasis in different cancers [37,38]. HTR2B has the highest RNA-seq logFC value in high-risk samples compared to low-risk ones. Moreover, low expression of the collagen type XI alpha 1 gene COL11A1 was correlated with poor survival [39]. Since there is a good correlation between expression and methylation, we tried a data fusion (DF) approach. Unfortunately, we have no methylation data for the Genoa dataset [40,41], and therefore we used a DF method that supported the integration of partial-omics data, NEMO [42], while previously applied methods such as jCMF or jSVD were not applicable [36,43]. NEMO produced 4 clusters with different survival curves whose differences were statistically significant (Appendix A). Class 1 and 4 are high-risk classes, while 2 and 3 have low risk (Figure 7): 5 metastatic patients were misclassified.

One patient with no feature, one with M3, 8q, BAP1, and one with only chr3 monosomy were classified in class 3, while two other metastatic patients were classified in class 2, one had 6p, SF3B1 mutation, the other also 8q (Table 1 and Table 2). These patients can be found in the right part of the heatmap of coupled and single genomic factors, distant from the leftmost part of the plot, where the majority of metastatic patients were clustered (Figure 3 and Figure 5). Hence, misclassification is probably due to their diversity in the expression profile compared to the majority of metastatic patients. Misclassification likely relies on the intrinsically stochastic process of metastasization [44].

## 3. Discussion

Uveal melanoma metastatic disease has a complex development: the loss of both BAP1 alleles with mutations and chromosome 3 deletion and/or chr8q gain could drive the formation of different metastases that, after colonization of the liver, can spread to different organs. Each tumoral cell line will randomly develop additional mutations and CNA to proliferate in different organs [16,45]. Metastases originate before the primary tumor is extracted and grow into clinically overt metastases only later on, when additional genomic events have accumulated in the cells of micro-metastases [16] or when a favorable microenvironment is created [46]. However, primary and metastatic tumors will continuously accumulate new CNA and mutations, increasing the genomic diversity between the different tumor subclones [23,45]. The most frequent CNA are 3p loss, 8q gain, and 6p gain: the first two are associated with bad prognosis and the third is considered to have a protective effect, possibly since M3 and 6p rarely occur in combination [25]. Other events such as loss of 1p and 16q are rare but found in most aggressive M3 UM tumors [3].

Chr3 monosomy, BAP1 mutations, and chr8 gain are the principal features of patients with metastases [3,10,13] as shown in Figure 1 and Figure 2. These events have previously been linked to liver metastases [47]. The current model of UM tumorigenesis is based on the Darwinian evolutionary selection of a sequence of genetic aberrations. Here we show that genetic and epigenetic alterations are independently acquired with each alteration increasing the metastatic risk. However, the alterations do not occur sequentially and metastasis is the consequence of an accumulation of randomly introduced high-risk features. This model also allows for secondary drivers that further derange the central oncogenic signaling pathway [5].

The rapid metastasizing capacity of uveal melanoma and the inability to improve the outcome despite progress in early diagnosis and almost perfect control of the primary disease also indicates that the progression of the disease is not gradual, but rather the result of divergent trajectories of the initially transformed uveal melanocytes [14]. In the classical gradualism model of tumor evolution, there is a period of latency between driver mutations and multiple, independent secondary hits during the transformation phase of tumorigenesis. This tumor evolution model has recently been challenged by a new model, the so-called “punctuated equilibrium” or “big bang” model that predicts an initial phase of great genomic instability (the cancer-initiating event) followed by the selection of clones carrying genetic alterations conferring selective advantage and capable of invasion and metastasis [31,48] that has also been proposed for UM [14]. Strong evidence indicates that the genetic alterations typical of uveal melanoma usually arise very early in tumor development followed by clonal stasis [49]. Given the evidence cited and the results of the present analysis (and despite the paucity of molecular lesions that characterize metastatic UM), UM metastasization fits better with the punctuated equilibrium model of cancer development than with the classical gradualism model. A big bang followed by the outgrowth of viable clones that might further be restricted by cellular competition leading to clonal extinction [32] seems to hold for many if not all tumors, even when the extension of the big bang can be very limited (or clonal extinction very extended) as in UM.

Our analysis shows the independence of the single dismal prognostic factors. This is in contrast with a model of stepwise acquisition of molecular lesions determining an increasing risk. Bakhoum and colleagues have performed single-cell transcriptomics on primary UM, and they show the composition of the tumors with various proportions of cells showing class 1 or class 2 gene expression or features of one of the four prognostic TCGA classes [14]. The authors interpret this finding as evidence of the definition of the risk being determined by the relative proportion of high and low-risk cells present in the tumor. In our interpretation, too few cases have been analyzed to establish a “quantitative” risk model. In any case, the cellular composition model is in stark contrast with the detection of drastically different clusters of low- and high-risk cases by bulk transcriptomics which we also confirm here. The cellular composition model would predict a continuous risk-related gene expression profile. We think that risk factors occur randomly and independently (except for M3 and BAP1 loss) and concur to define the risk. To establish whether and to which extent the risk might also depend on quantitative aspects of different cell populations, many more tumors must be analyzed. Yet we are inclined to predict that such analyses will dismiss the cellular composition model. Cell morphology is associated with risk in a non-quantitative manner since the mere presence of epithelioid cells rather than the ratios of epithelioid and spindle cells is associated with risk [50]. For certain, further genomic analyses are required to investigate the dynamics of UM metastasis and tumor heterogeneity.

## 4. Materials and Methods

### 4.1. Data Source

We obtained CNA data (3p, 6p, 8q) and gene expression profiles of UM tumors from 113 patients with known gene mutation profiles. The dataset is composed of 80 RNA-seq expression profiles from TCGA [13], obtained from UCSC-Xena Browser (http://xena.ucsc.edu/ (accessed on 27 September 2023).level 3 data, log2(x + 1) transformed RNA-Seq by Expectation Maximization, RSEM, normalized counts) and 33 Affymetrix arrays from our previously published works (GEO: GSE27831, GSE51880) [40,41] (Table 3).

### 4.2. UM Gene Expression Data Collection

The two datasets have different proportions of metastatic patients, more abundant in the Genoa dataset compared to the TCGA one, this fact is in part reflected by the shorter follow-up time in the first dataset. Monosomy of chromosome 3 and 8q gain is observed in the majority of patients in both datasets. The uveal melanoma gene expression dataset file has been prepared following the procedure published by Piaggio et al. [12]. Briefly, microarray probes were collapsed to gene symbols to the maximum variance dataset with the weighted correlation network analysis (WGCNA) R package [51]. Expression profiles were merged in a single file, without batch effects, using the Combating Batch Effects When Combining Batches of Gene Expression Microarray Data (COMBAT) algorithm, as implemented in the inSilicoMerging R package [52]. Clinical and molecular data of the Uveal melanoma expression dataset were collected from the original publications.

### 4.3. UM Expression Data Analysis

Differentially expressed genes were computed with the Limma R package [53]. Each group was compared with the remaining samples with a couple of genomic features, such as BAP1/M3, 8q|M3, 8q|SF3B1, 6p|SF3B1, 6p|8q, M3|SF3B1. Survival curves were computed with the survival R package, considering metastasis development and patient death for causes different from Uveal Melanoma as censored [54,55]. Gene ontology enrichment has been computed with Clusterprofiler on DEG for which at least one comparison had an absolute value of logFC greater or equal to two [56,57]. We used the NEMO R package [42], selecting k = 4, to perform Data Fusion on the Uveal Melanoma dataset. We considered the whole expression dataset and only TCGA data for the methylation domain. Both matrices only had differentially expressed genes for a couple of factors, with at least one logFC ≥ 2. Venn diagrams were created with the ggvenn R library [58], cowplot R package was used to join multiple Venn plots in one figure [59].

### 4.4. Statistical Analysis

Only significant results (method’s *p*-value < 0.05) were considered in the analysis computed with Limma and Clusterprofiler R package. The relationship between gene expression and mean methylation levels was computed in R, using the Pearson Correlation Coefficient (PCC), associated *p*-value was computed with the cor.test R function, all values are reported in Appendix A. Significance of survival curves was computed with the Log-rank test, using the surv_pvalue function of the Survminer R package [54,55], *p*-values are reported in Appendix A.

## 5. Conclusions

Despite success in identifying the canonical genomic alteration in UM, how and when these events arise during tumor evolution remains unknown [49]. Our results show that the genomic aberrations usually arise in an early punctuated burst, followed by neutral evolution, indicating that CNA in the UM tumorigenesis is neither gradual nor follows multistep carcinogenesis. This implies that the metastatic potential of UM is set in stone early in tumor evolution and may explain the striking differences between uveal melanoma and cutaneous melanoma responses to treatment and, last but not least, the imbalance between advances in primary treatment and the lack of advance in the treatment of the metastatic disease. These findings challenge the current model of UM progression and provide insight into the mutational process that gives rise to metastatic uveal melanoma.

The CNA landscape of 921 UM samples in the dataset presented by Lalonde et al. [3] shows that most metastatic patients have chr3 monosomy and 8q gain (Figure 8) as observed in the UM expression dataset considered in our work (Figure 1 and Figure 2). A sequentiality between CNA events is not the rule in UM evolution different CNA events can develop without a fixed order (e.g., patients with 6p or 8q amplifications without M3).

## Figures and Tables

**Figure 1 ijms-24-15602-f001:**
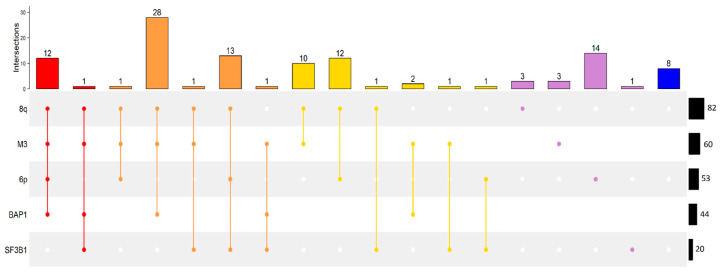
Upset plot of all patients in Genoa and TCGA UVM dataset. Each row represents a genomic alteration: chromosome 8q gain (8q), monosomy of chromosome 3 (M3), chromosome 6p gain (6p), BAP1 and SF3B1 mutation. Each bar represents the number of patients that have a set of genomic factors, represented as points connected by a line (e.g., 12 patients have 8q, M3, 6p, and BAP1). Black bars, on the right, are the total number of patients in the dataset with a specific genomic alteration (e.g., 82 patients have 8q). Bar colors are related to the number of genomic events in the same patient: 4 = red, 3 = orange, 2 = gold, 1 = violet, 0 = blue.

**Figure 2 ijms-24-15602-f002:**
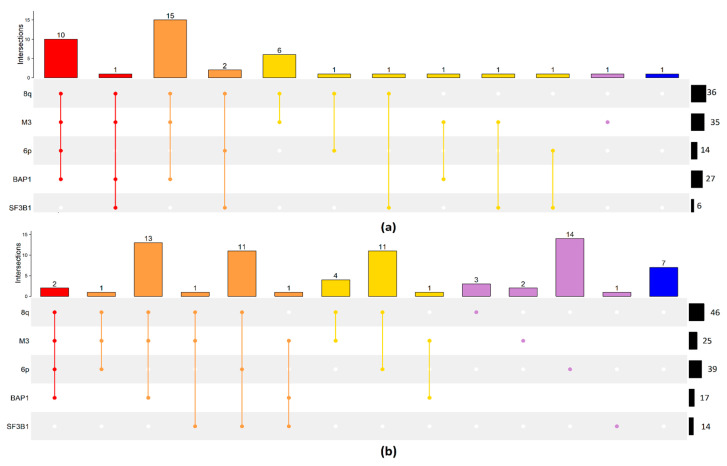
(**a**,**b**). Upset plot of all samples of primary UM that developed metastasis within the time of follow-up (**a**), the remaining are reported in panel (**b**). Each row represents a single genomic alteration: chromosome 8q gain (8q), monosomy of chromosome 3 (M3), chromosome 6p gain (6p), BAP1 and SF3B1 mutation. Each bar represents the number of patients that have a set of genomic factors, represented as points connected by a line (e.g., 12 patients have 8q, M3, 6p, and BAP1). Black horizontal bars, on the right, are the total number of patients in the dataset with a specific genomic alteration (e.g., 82 patients have 8q). Bar colors are related to the number of genomic events in the same patient: 4 = red, 3 = orange, 2 = gold, 1 = violet, 0 = blue.

**Figure 3 ijms-24-15602-f003:**
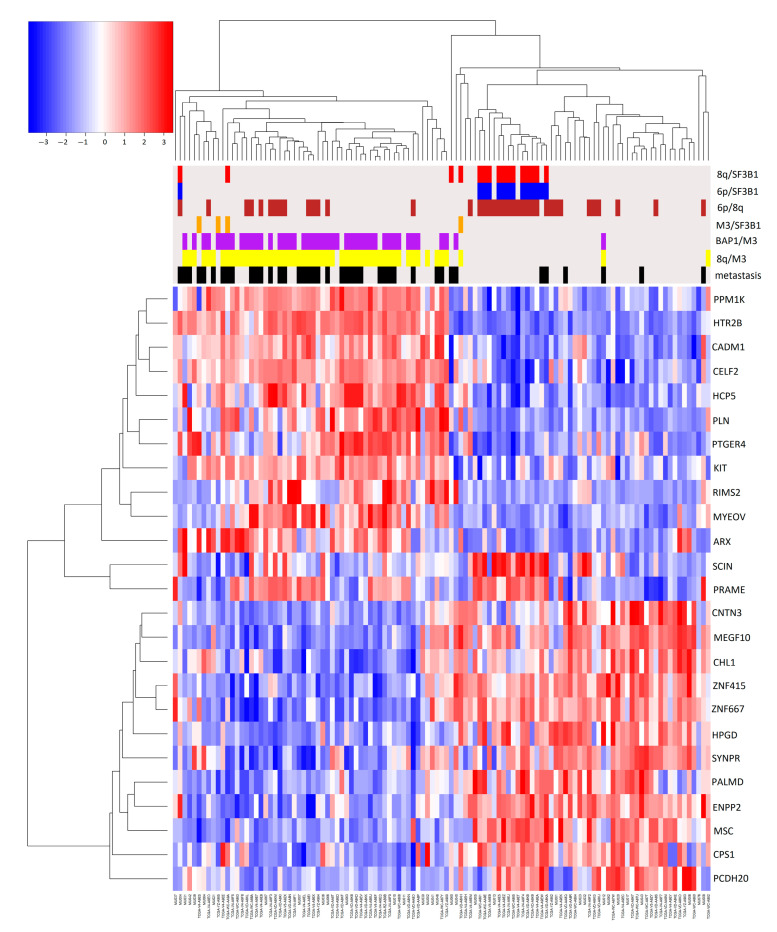
Heatmap of differentially expressed genes. In this heatmap, we reported genes that were differentially expressed comparing one couple of factors vs. all (e.g., patients with 8q and M3, 8q/M3, vs. all). In this heatmap only differentially expressed in more than one group, with at least one comparison with logFC greater or equal to 2, are reported. Heatmap colors represent the expression values relative to the mean value: red indicates expression higher and blue lower than mean, white at mean. The color intensity indicates the distance from the mean value. On the top of the heatmap combination of genomic events are reported with different colors: 8q/SF3B1 = red, 6p/SF3B1 = blue, 6p/8q = brown, M3/SF3B1 = orange, BAP1/M3 = purple, 8q/M3 = yellow. Patients that developed metastasis within the time of follow-up = black.

**Figure 4 ijms-24-15602-f004:**
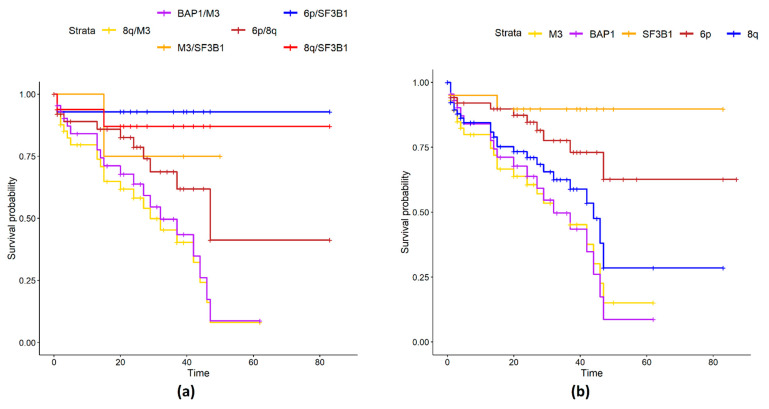
(**a**) Survival curves of couples of genomic factors. The survival of patients with two genomic factors together is reported (e.g., the curve of patients with 8q and M3 is reported in yellow). (**b**) Survival curves of single genomic factors. The survival of patients with one genomic factor together is reported (e.g., patients that have M3 are reported in yellow).

**Figure 5 ijms-24-15602-f005:**
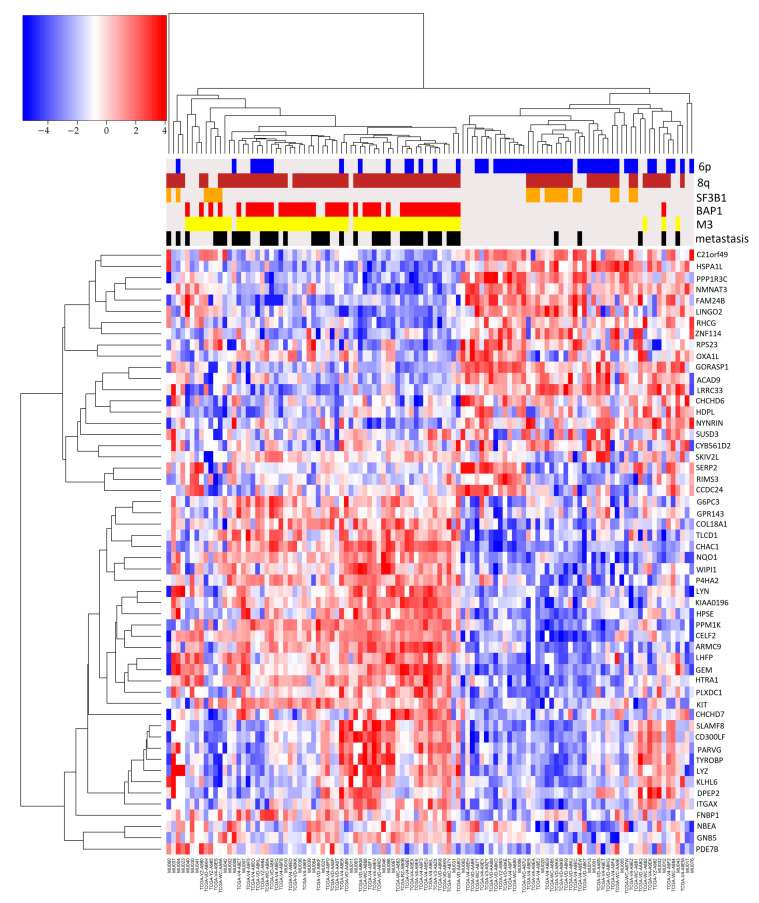
Heatmap of differentially expressed genes in single genomic factors. In this heatmap, we reported genes that were differentially expressed by comparing one factor vs. all (e.g., patients with 8q vs. all patients without 8q gain). In this heatmap only differentially expressed genes in more than one group, with at least one comparison with logFC greater or equal to 2, are reported. Heatmap colors represent the expression values relative to the mean value: red indicates expression higher and blue lower than mean, white at mean. The color intensity indicates the distance from the mean value. On the top of the heatmap genomic events are reported with different colors: 6p = blue, 8q = brown, SF3B1 = orange, BAP1 = red, M3 = yellow. Patients that developed metastasis within the time of follow-up = black.

**Figure 6 ijms-24-15602-f006:**
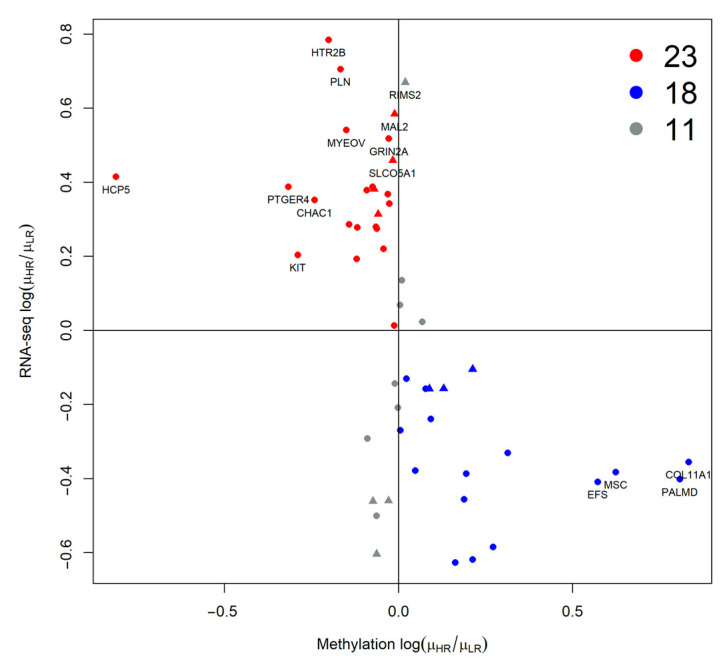
Scatterplot representing log rate of the mean RNA gene expression and methylation data of high-risk (3, 4) over low-risk (1, 2) patients of genes on Appendix A. Overexpressed and down-methylated genes are reported in red, while blue color is used for downregulated and up-methylated genes. All other points are colored gray. Genes overlapping on CNA regions are represented as triangles.

**Figure 7 ijms-24-15602-f007:**
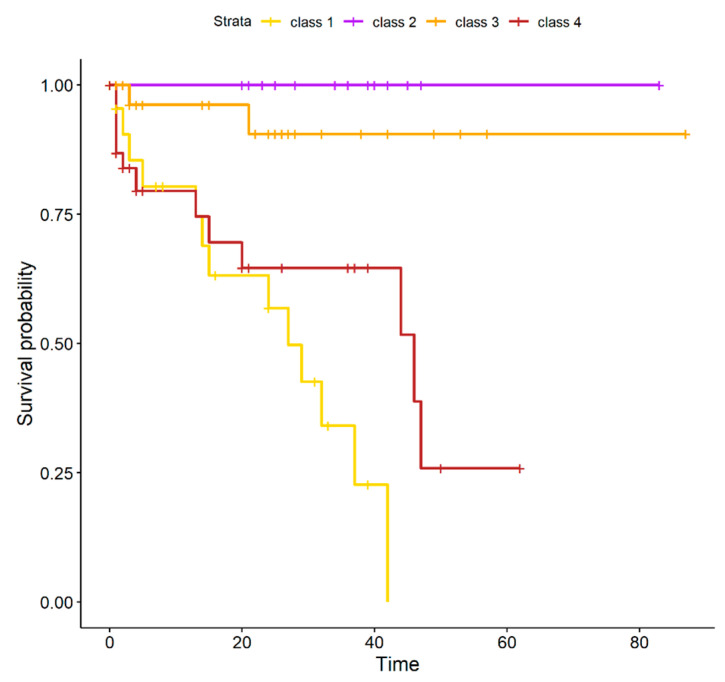
Survival curves as predicted by Data Fusion. The survival curves of the classes were found by data fusion of the whole UM RNA expression dataset and TCGA methylation data on DE genes of couples of factors with at least one comparison with logFC ≥ 2. In yellow is reported the survival of class 1 patients. Class 1 and 4 have a high risk of metastasis, while for Class 2 and 3, the risk is lower.

**Figure 8 ijms-24-15602-f008:**
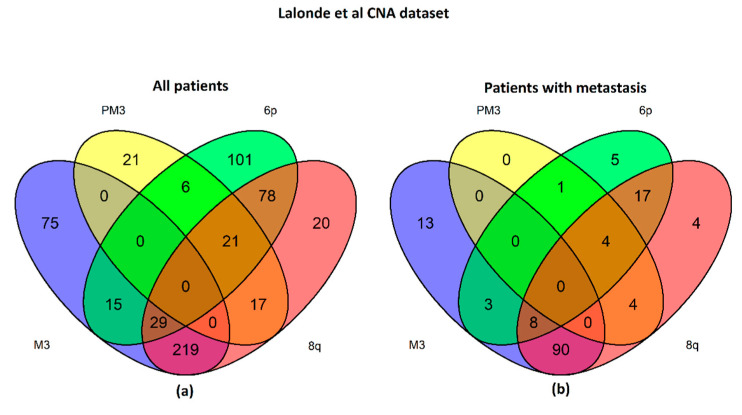
Analysis of CNA detected in UM samples, as reported by Lalonde et al. in Supplementary Table S1 of [3]. The two Venn ((**a**), all patients; (**b**), patients with metastasis) plots show a set of molecular features in the whole dataset or in patients that developed metastasis during follow-up: chromosome 3 monosomy (M3, blue), partial monosomy (PM3, yellow), gain on chromosome 6p (green) or 8q (red).

**Table 1 ijms-24-15602-t001:** Features of patients in clusters defined by NEMO.

Cluster	M3/BAP1	Metastasis	All
class 1	24	16	25
class 2	0	2	18
class 3	2	3	31
class 4	34	20	39

**Table 2 ijms-24-15602-t002:** Features of metastatic patients classified in clusters 2 and 3.

Patients_ID	Metastasis	M3	6p	8q	BAP1	SF3B1	Cluster
TCGA-V4-A9EW	1	0	1	0	0	1	2
TCGA-VD-A8KA	1	0	1	1	0	1	2
MU012	1	1	0	1	1	0	3
MU043	1	1	0	0	0	0	3
TCGA-VD-AA8Q	1	0	0	0	0	0	3

**Table 3 ijms-24-15602-t003:** Clinical and molecular information of patients collected in the Genoa and TCGA Uveal melanoma gene expression dataset.

Parameter	Category	Genoa Dataset (33)	TCGA Dataset (80)
Gender	Male/Female	19, 14	45, 35
Age	Years	65 (42–83)	62 (22–86)
Metastases	Yes	15	26
	No	18	54
	Alive	21	56
State	Dead (metastatic disease)	8	20
	Dead (other)	4	4
Follow up	Months	2 (0–5)	27 (0–87)
Chromosome 3p	loss	18	42
Chromosome 8q	gain	23	59
Chromosome 6p	gain	8	45
BAP1	mutation	9	35
SF3B1	mutation	2	18

## Data Availability

Uveal melanoma gene expression, and methylation data used in this work is publicly available. TCGA RNA-seq, methylation data is available on http://gdac.broadinstitute.org/ (accessed on 27 September 2023), Affymetrix arrays were previously published on GEO (GSE51880, GSE27831).

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
