# Peer review of "Interdependence of Molecular Lesions That Drive Uveal Melanoma Metastasis"

_ijms, 2023, doi:10.3390/ijms242115602_

Round 1
Reviewer 1 Report
Comments and Suggestions for Authors
The manuscript entitled “Interdependence of Molecular Lesions that Drive Uveal Melanoma Metastasis” describes several gene expression events that show a non-significant association with outcome of uveal melanoma (UM). Authors aimed to identify the molecular basis underlying UM evolution and metastasis.
The work is comprehensive, well written, and meets the stated objectives. The introduction is highly detailed, the discussion is comprehensive, but there are some minor issues to be improved in the Methods and Results sections:
1. The legend in Figure 1 uses a different abbreviation than the figure itself, for example CNA8q vs. 8q. Please use the same designation in the text.
2. Figure 1: There is no explanation for the colours of the bars.
3. Figure 2 legend have to be complemented with more descriptions as Figure 1.
4. The manuscript lacks statistics; for methods, the Statistics section is needed. For example, correlation and survival were calculated, but described. Were data normally distributed? Or how was the relationship between gene expression and methylation data calculated? What was the p-value of significance?
5. Figure 3. and Figure 5: The colours of the heat map should be explained.
6. The result of the heat map (Figure 3) is not described clearly in the text. The results of this analysis starts in 2.1. section, then continue in 2.2., which causes confusion in interpretation, authors should organize the description of this analysis in one section.
Author Response
We would like to thank Reviewer 1 for his useful comments that helped to improve our manuscript. We modified the main text addressing all points requested. Briefly, we modified the main text and we added a section on statistical methods, we reported p-values for all survival curves in supplementary table 2.
Reviewer 2 Report
Comments and Suggestions for Authors
The manuscript authored by Reggiani et al., titled "Interdependence of Molecular Lesions that Drive Uveal Melanoma Metastasis," employs a robust analysis method that integrates diverse datasets and molecular profiling to investigate uveal melanoma (UM) metastatic progression. The findings reveal that the metastatic risk in UM is influenced by a limited set of molecular lesions, including somatic mutations (SF3B1 and BAP1) and copy number alterations (CNA, monosomy of chromosome 3 [M3], chr8q gain, chr6p gain), although the precise sequence of events remains unclear. The study's comprehensive approach, which draws from multiple datasets, confirms a significant association between BAP1 mutations and M3 in high-risk patients. Importantly, it highlights the independent occurrence of other risk factors (6p, 8q, M3, SF3B1 mutation), challenging the notion of a predefined sequence in UM metastasis. Hierarchical clustering of gene expression data further emphasizes the interplay of these risk factors, shedding light on their additive effects. This manuscript offers valuable insights into UM metastasis, emphasizing the complexity of its molecular. It is an important contribution to the field of ocular oncology and holds significance for future research and clinical applications. The quality and significance of this study merit its acceptance for publication.
Author Response
We would like to thank Reviewer 2 for the revision work and interest in our paper.